# The Declining Trend in Adolescent Drinking: Do Volume and Drinking Pattern Go Hand in Hand?

**DOI:** 10.3390/ijerph19137965

**Published:** 2022-06-29

**Authors:** Ingeborg Rossow, Elin K. Bye, Inger Synnøve Moan

**Affiliations:** Department of Alcohol, Tobacco and Drugs, Norwegian Institute of Public Health, 0213 Oslo, Norway; elinkristin.bye@fhi.no (E.K.B.); ingersynnove.moan@fhi.no (I.S.M.)

**Keywords:** alcohol consumption, heavy episodic drinking, drinking culture, adolescents, secular trends

## Abstract

Traditionally, adolescent drinking cultures differed between Nordic and Mediterranean countries; the former being characterised by low volume and relatively frequent heavy episodic drinking (HED). Across these drinking cultures, we examined the associations between alcohol volume and HED with respect to (i) secular trends at the country level and (ii) individual-level associations over time. The data stem from the European School Survey Project on Alcohol and Other Drugs (ESPAD) conducted among 15–16-year-olds in Finland, Iceland, Norway, Sweden, France and Italy, employing six cross-sectional surveys from 1999 to 2019 (n = 126,126). Both consumption volume and HED frequency decreased in all Nordic countries and displayed a curvilinear trend in France and Italy. In all countries, consumption volume and HED correlated highly over time at the country level. At the individual level, the correlation was positive but with a varying magnitude over time and between countries. In 1999/2003, the alcohol volume–HED correlation was significantly higher in the Nordic compared to the Mediterranean countries but became significantly weaker in Finland, Norway and Sweden and remained stable in France, Iceland and Italy during the period. In conclusion, while trends in consumption volume and drinking patterns went hand in hand at the aggregate level, the association at the individual level weakened over time in several Nordic countries, along with the substantial decline in adolescent drinking since 2000.

## 1. Introduction

Alcohol consumption is a significant and modifiable risk factor for health and social harms [1], and alcohol consumption accounts for a particularly large fraction of premature mortality in the age group 15–49 years [2]. There are two aspects of alcohol consumption that are essential for risk of health harm or social harm; (i) the overall amount of alcohol consumption and (ii) the drinking pattern, in particular the frequency of heavy episodic drinking (HED) [1]. Thus, factors that impact either of these, including effective policy measures, are of utmost importance for public health. While there is strong evidence that overall consumption can be reduced effectively by control policy measures, including high taxes and restrictions on physical availability [3], it is less clear whether these measures or other factors are effective in changing the drinking pattern in a population. For instance, it is often assumed that an intoxication-oriented drinking pattern is deeply culturally rooted and, to a small extent, influenced by control policies or other societal factors [4], as outlined in the following. 

In European countries, total consumption in the adult population varies substantially, both over time and between countries [5]. Drinking patterns, on the other hand, seem to have been fairly stable over time and are apparently more strongly culturally rooted and more resistant to various changes in society. For example, drinking patterns in the Nordic countries have traditionally been characterised by an intoxication-oriented culture with relatively frequent HED occasions despite low total consumption. This typology of drinking culture in the Nordic countries stands in contrast to that of southern European countries, including Italy and France, where total consumption has been higher, whereas the drinking culture is less intoxication-oriented, and HED tends to occur less frequent [6]. Thus, at the population level, across countries and populations, the variation in consumption volume is not necessarily reflected in patterns of HED. In other words, a high consumption volume does not necessarily go hand in hand with a high level of HED. From a global perspective, there is no significant correlation between overall consumption and HED prevalence (our calculations from data in Table 13.5 in Gmel et al. [7]). For example, South East Asian countries have twice the proportion of drinkers with weekly HED compared to European countries, although the consumption volume per drinker is quite similar [7]. 

It is also found that consumption volume and HED frequency display different patterns across socio-demographic groups within populations. For instance, volume of consumption often peaks in middle age and increases with socioeconomic status (SES), whereas HED frequency typically peaks in young adulthood and tends to decrease with increasing SES [8]. These observations could suggest that—both between and within populations—the volume of consumption and HED are separate dimensions of drinking behaviour concerning how various social factors influence them. Here, we will examine this issue further by exploring whether or to what extent the recent decline in alcohol consumption among adolescents was accompanied by a change in drinking pattern. 

A growing amount of literature shows that over the past two decades, the prevalence of drinking and the overall amount of drinking have declined among adolescents in many European countries [9,10,11,12,13,14,15,16,17,18,19]. Judging from data from the European School Survey Project on Alcohol and other Drugs (ESPAD), the decline appears to be particularly steep in Nordic countries, especially in Finland, Iceland, Norway and Sweden, whereas less or no decline is observed in many Mediterranean countries [20]. In other words, a substantial decline in consumption occurred, particularly in countries with an intoxication-oriented drinking culture. Yet, to this end, it is not clear whether a decline in HED accompanied the decline in adolescents’ consumption. There are some examples that a decline in drinking prevalence was accompanied by a decline in HED prevalence from one survey to the next [10,21], although there are also several examples of the opposite [10]. Moreover, it is not clear whether any association in trends in volume and drinking pattern is moderated by a HED-oriented drinking culture. Hence, the decline in alcohol consumption over time among adolescents provides an opportunity to assess whether drinking pattern, in terms of HED frequency, tends to go hand in hand with consumption. In this paper, we will examine this issue empirically.

By examining the alcohol volume—HED association in populations from a time dynamic perspective, we empirically test the idea that the drinking pattern is relatively stable within a population irrespective of changes in the overall consumption volume. Specifically, we asked: (1)Are country-level changes in the volume of alcohol consumption associated with country-level changes in HED?(2)Are the individual-level associations between the volume of consumption and HED different between and within countries with different levels of HED?

## 2. Material and Methods

Data stem from the European School Survey Project on Alcohol and Other Drugs (ESPAD) trend database. The ESPAD survey was conducted among samples of 10th graders in nearly all European countries at four-year intervals from 1995 to 2019 [22]. Identical (or comparable) questions on drinking behaviour have been applied in all countries at all survey waves, which provided an opportunity to examine our research questions. In these surveys, self-reported alcohol consumption refers to the intake of alcoholic beverages, including beer, alcopops/premixed drinks, wine and spirits [22]. The present analysis was based on six waves of the ESPAD survey (1999, 2003, 2007, 2011, 2015, and 2019). We included data from four Nordic countries with a HED-oriented drinking culture (Finland, Iceland, Norway and Sweden) and from two Mediterranean countries where drinking is typically less HED-oriented (France and Italy). The total sample size across all surveys and countries comprised n = 126,126 students aged 15–16. 

### 2.1. Measures 

We applied a semi-continuous measure of drinking frequency. The students were asked: “On how many occasions (if any) have you had any alcohol beverage to drink during the past 30 days?”, with response alternatives: ‘0’, ‘1–2’, ‘3–5’, ‘6–9’, ‘10–19’, ‘20–39’ and ‘40 or more’. The responses were recoded to the number of days per month; 0, 1.5, 4, 7.5, 14.5, 30 and 30, respectively. Notably, the response category ‘40 or more’ was coded 30 because it is the maximum possible number of days in a 30-day period; however, less than 1% of the respondents ticked off this response option. The volume of consumption past 30 days was calculated by multiplying the measure of drinking frequency past 30 days and the amount of alcohol consumed on the most recent drinking occasion. The latter was obtained from beverage-specific questions on volume consumed on the most recent drinking occasion; that is, the students were asked: “If you drank beer/cider/premixed drinks/wine/spirits that last day you drank any alcohol, how much did you drink?”. The response categories (ranging from none to >200 cl for beer, cider and premixed drinks, up to >74 cl for wine, and up to >24 cl for spirits) were calculated into centiliters of pure alcohol, using the midpoints of the categories (e.g., 50–100 cl = 75 cl) and the following alcohol content for the various beverages: beer/cider, 5%; premixed drinks, 4.5%; wine, 12%; spirit, 38% [19]. Thus, the volume of alcohol consumption was measured in centiliters of pure alcohol past 30 days. This procedure has also been applied in previous studies employing ESPAD data [19]. Notably, in France, for the survey year 2003, beer was not included, and for the survey year 2011, wine was not included among the questions on volume at the most recent drinking occasion. Thus, to obtain comparable data at the aggregate level, we imputed volume data for these two survey years, assuming that beer accounted for the same proportion of total volume in 2003 as in the two neighbouring survey years (i.e., 1999 and 2007), and a corresponding assumption for wine. 

The frequency of heavy episodic drinking (HED) was assessed by asking how many times in the past 30 days the students had drunk a certain number of drinks or more on one occasion. This corresponded to five or six drinks, depending on country-specific container sizes and average alcohol content per beverage. Response options were ‘none’, ‘1 time’, ‘2 times’, ‘3–5 times’, ‘6–9 times’, ‘10 or more times’, and by employing mid-point values, these response categories took the values ‘0’, ‘1’, ‘2’, ‘4’, ‘7.5’, and ‘12’, respectively. For some respondents who reported drinking frequency ‘1–2 times’ and HED frequency ‘2 times’, the latter took the value ‘1.5 times’. 

We applied an indicator of HED-oriented drinking pattern by calculating the ratio of HED frequency: to drinking frequency (i.e., the number of HED occasions past 30 days divided by the number of drinking occasions past 30 days among drinkers only). Thus, the ratio is equivalent to the proportion of drinking occasions that were HED occasions, and the larger the ratio, the more HED-oriented drinking pattern. 

### 2.2. Analyses

We analysed the data both at the aggregate level and at the individual level. At the aggregate level, we examined co-variation in trends for volume and drinking pattern, thereby reflecting trends in drinking prevalence. Hence, in these analyses, we included all students. At the individual level, we examined the extent to which volume and drinking pattern (i.e., HED frequency) co-varied among drinkers and whether the magnitude of association changed over time, and in these analyses, we included only past 30 days drinkers (i.e., 45.5% of the sample, n = 56,470). 

Thus, we first examined the aggregate-level associations between the mean volume of alcohol consumption and mean number of HED occasions, for which each survey year represents a data point. Trends in consumption volume and HED frequency were depicted for each country and survey year in graphs for visual inspection, and bi-variate associations were estimated employing Pearson’s correlation coefficient. In these analyses, we also applied the measure of drinking frequency (described above) as an indicator—or proxy measure—of consumption volume. The reasons for this are that drinking frequency is inherently and closely correlated with the overall volume of consumption, and the reliability of the estimated volume at the most recent drinking occasion may differ systematically with drinking cultures across countries and over time. Regarding the latter, the reliability of reported consumption is better when the recall period is short [23]. Thus, for those who drink infrequently, the recall period for the most recent drinking episode is on average longer than for those who drink frequently. Hence, for infrequent drinkers, the reported volume at the most recent drinking episode is likely less reliable. 

Next, we examined the individual-level association between consumption volume and drinking patterns among drinkers only. For these analyses, we applied two types of analyses of association for each country. First, we estimated the proportion of drinking occasions that were HED occasions for each survey year. Next, we regressed HED frequency on drinking frequency (as a proxy measure for volume) in linear regression models and thereby estimated how much a change in drinking frequency predicted a change in HED frequency. In these analyses, we applied pairs of survey years (i.e., 1999/2003, 2007/2011 and 2015/2019) to increase robustness when examining whether these associations changed over time within each country. Hence, we tested differences between each country’s estimated regression coefficients at the two endpoints of the observation period (1999/2003 versus 2015/2019). Furthermore, we tested whether the regression coefficients differed between countries at the two endpoints of the observation period. Regression coefficients with non-overlapping 95% confidence intervals were considered statistically significantly different (*p* < 0.05) [24]. 

The number of missing responses varied across the country, survey year and variable. We employed weighted data in all analyses for those countries that provided sampling weights (all but Iceland and Sweden). Weights were calculated to account for gender, the geographical distribution of the target population, type and size of schools and immigrant background (only Finland). We employed SPSS (version 27, IBM, Chicago, IL, USA) for the statistical analyses.

## 3. Results

We first examined the aggregate level data.

Table 1 shows substantial differences (non-overlapping confidence intervals) between countries for drinking frequency, volume and drinking pattern. The Nordic countries had low drinking frequency and volume levels compared to the Mediterranean countries. While average HED frequency did not differ much across countries, the relative importance of HED—as expressed in the ratio figure—was higher in the Nordic countries compared to the Mediterranean countries. 

Table 2 shows the proportion of drinkers past 30 days by survey year and country. In all the four Nordic countries, the proportion of drinkers decreased substantially over the 20 years and was halved or more. In the two Mediterranean countries, the proportion was higher in 2007 and 2001 compared to the two preceding and the two subsequent survey years. 

Figure 1 shows substantial differences between countries with respect to both levels and trends in drinking frequency. The Nordic countries (Iceland, Finland, Norway and Sweden) displayed both low levels and overall decreasing trends. The Mediterranean countries (France and Italy), on the other hand, displayed higher levels at all points in time and a curvilinear trend with a peak in the middle of the period, resembling a reversed U-shaped curve.

Figure 2 shows data for the past 30 days’ consumption volume with a fairly similar pattern to that observed in Figure 1: both the levels and the trends by country display a very similar pattern.

Figure 3 shows a declining trend in HED frequency in the Nordic countries and a curvilinear trend with a peak in 2007 in the Mediterranean countries. 

Thus, from visual inspection of the diagrams, we observe a roughly similar patterning of trends in drinking frequency, volume and HED frequency in each country and the two groups of countries by drinking culture: a downward trend in the four Nordic countries and a peak in the middle of the period in the Mediterranean countries. 

We estimated the associations between the three aggregate measures of drinking frequency, volume and HED frequency by country, as reported in Table 3. For all countries, there was a high correlation coefficient between drinking frequency and HED frequency; this was also the case for the correlation between consumption volume and HED frequency. 

We further examined the HED/drinking frequency ratio by year and country, as depicted in Figure 4. 

Throughout the whole period, this ratio was higher in the Nordic countries compared to the Mediterranean countries, and we observed a slightly decreasing trend in the Nordic countries over time compared to the other countries. 

Next, we explored associations between consumption volume and HED at the individual level and among past 30 days drinkers only. Figure 5 shows the average proportion of drinking occasions that were HED by survey year and country. While the proportion remained relatively low and stable over the observation period in France and Italy, the proportion tended to be higher, yet with a decreasing trend in the Nordic countries.

Moreover, we estimated to what extent drinking frequency was associated with HED frequency, and we regressed HED frequency on drinking frequency for pairs of years and each country. The associations as linear regression coefficients are displayed in Table 4. In 1999/2003, the association was significantly stronger (i.e., larger regression coefficient) in the Nordic countries compared to the Mediterranean countries, as 95% confidence intervals do not overlap for any pair of Nordic-Mediterranean countries (Table 4). In three of the Nordic countries, i.e., Finland, Norway and Sweden, the drinking frequency-HED association was significantly weakened from 1999/2003 to 2015/2019, whereas in Iceland and the Mediterranean countries, the association remained at approximately the same level throughout the observation period (Table 4, Figure 6).

Hence, the north–south gradient in the magnitude of the drinking frequency–HED association was less clear at the end of the period than at the start. 

## 4. Discussion

Alcohol consumption is a major risk factor for morbidity and premature mortality among young people [2], and a HED-oriented drinking pattern particularly increases the risk of injuries and other acute harms [1]. In this study, we explored whether, or to what extent, similar movements in heavy episodic drinking accompanied trends in adolescent alcohol consumption volumes. We analysed data from countries representing two classic typologies of drinking cultures: the Nordic countries representing an intoxication-prone culture with a low consumption level, and the Mediterranean countries, representing a moderation-prone culture with a high consumption level. We found that consumption volume and drinking pattern were highly correlated over time in all countries at the aggregate level. Thus, whether consumption declined steadily, as in the Nordic countries, or increased and thereafter decreased, as in the Mediterranean countries, the extent of heavy episodic drinking followed ‘hand-in-hand’. At the individual level, among drinkers, we also found a substantial association between the two; however, the magnitude of the association differed both between countries and within countries over time. Hence, the clear differences between Nordic and Mediterranean countries found at the start of the observation period became smaller over the two decades from 1999 to 2019. While the volume-HED association was relatively weak and stable throughout the period in the Mediterranean countries, it weakened over time in three of the Nordic countries. Thus, overall, the aggregate-level findings suggest that trends in consumption volume are accompanied by very similar trends in HED frequency, whereas individual-level findings suggest that among drinkers, the HED-oriented drinking culture was less stable in the Nordic countries. Moreover, the HED-oriented drinking culture tended to weaken over the two decades, along with the decline in consumption volume. 

We will offer some possible explanations for the observed findings in the following sections. First, within each country, we found that among all students, including non-drinkers, the secular trends in consumption volume and HED frequency were highly correlated, irrespective of drinking culture or consumption trends. Obviously, these trends are influenced by changes in the prevalence of drinking, which will impact the correlation between volume and drinking pattern. Hence, the individual-level analyses of drinkers only add to and nuance this picture. In three of the Nordic countries, the association weakened over time, implying that drinking occasions, to a lesser extent, involved HED. This suggests that the drinking culture among drinkers had changed from HED-oriented to a more moderation-oriented way of drinking. It is this latter observation that intuitively contradicts the idea about a strong, culturally rooted drinking pattern which is highly robust and fairly insensitive to influences over time. 

So, how can we explain these apparent changes in drinking culture in Nordic countries? Let us first start by rephrasing the question and ask: why did HED occasions decline relatively more than drinking occasions? Some tentative explanations can be found in recent qualitative studies of adolescent drinking and drinking patterns [25,26,27,28,29]. In a study from Norway, adolescents aged 15–16 years gave their accounts of reasons for non-drinking. Among these were the desire to avoid the negative consequences of drinking. One example of such consequences is having photos or videos of drunkenness and regrettable behaviour distributed on social media. Another example is the effects of drinking on achievements in sports or education [25]. A longitudinal study among 15–17-year-olds in Norway [27] found that adolescent drinkers highlighted the importance of responsible drinking and having control. Similarly, a qualitative study from Sweden conducted among somewhat older participants, i.e., 17–21-year-olds, showed that both abstainers and drinkers considered acceptable alcohol use as drinking moderate and being in control [28]. Another qualitative study from Sweden, based on interviews with participants aged 15–23 [29], found that young people emphasised having self-control and implementing self-improvement as important elements in becoming a successful person, for instance, in relation to school and the labour market. The authors argued that these accounts may explain why adolescents, to an increasing extent, postpone their drinking to the legal age and that they prefer to drink moderately when they decide to enter drinking settings [29]. Thus, it may be conceived that some of the driving forces behind the decline in consumption in the Nordic countries implied a somewhat stronger impact on HED in terms of avoiding harm from HED as reflected in the findings from the qualitative studies above and our findings showing steeper declines in HED frequency (Figure 3) than in drinking frequency (Figure 1) among adolescents in the Nordic countries. 

While, as noted above, many studies have examined possible explanations for the declining trends in adolescent drinking in the Nordic countries, less is known about likely explanations for changes in drinking behaviour among youth in Mediterranean countries. Two recent studies, employing data from many European countries, including France and Italy, found that neither computer-based activities nor parental control and support could explain population-level trends in adolescent drinking [30,31]. Thus, it remains unclear why the trends in adolescent drinking differ between Nordic countries and Mediterranean countries. Moreover, there is, to our knowledge, no relevant literature to suggest whether the tentative explanations for the relatively stronger decline in HED in the Nordic countries also apply to Mediterranean countries. 

Policymakers and the industry often state their interest in changing the drinking culture by reducing HED without reducing total alcohol consumption. Such ideas have been referred to as the “dream of a better society” [32] and “dream of a better order.” [33]. Our study suggests that in several countries, volume and drinking pattern went ‘hand-in-hand’ over time, whereas in a couple of countries, the drinking culture (in terms of the relative importance of HED to overall consumption) changed in a favourable direction over time. A key question then is, of course, whether such change can be achieved by targeted actions? We cannot speculate from the basis of the present findings, and further studies are clearly needed. 

Some study limitations should be noted. The ESPAD data on alcohol consumption stem from self-reports, and the volume of reported alcohol consumed is likely underestimated both at high levels [34] and among infrequent drinkers [35]. Correspondingly, HED frequency is likely under-reported as well. In addition, heavier drinkers are, compared to others, less likely to participate in alcohol surveys [36]. Taken together, this suggests that the overall level of consumption and heavy episodic drinking are likely downward biased. However, assuming that under-reporting affects all surveys equally and does not change considerably over time, its impact on country comparisons and temporal trends was considered negligible. Moreover, a downward bias in the level of consumption and HED frequency does not necessarily imply that the estimated *associations* between the two are substantially biased. Another issue pertains to the measure of alcohol quantity consumed at the last drinking occasion to calculate consumption volume. While this measure, as compared to the usual quantity per occasion, is considered less subject to under-reporting [37], it may pose a problem in our context, where we compare consumption volume across different drinking cultures. Specifically, at the population level, the amount consumed at the most recent drinking occasion should ideally reflect the distribution of light and heavy drinking occasions. It is, however, possible that the extent to which this distribution is well captured by using quantity at the most recent occasion may vary across drinking cultures. For instance, it is conceivable that in drinking cultures with infrequent drinking, including the Nordic countries, the heavier drinking occasions may be over-represented by using the ‘most recent occasion’ measure, whereas, in other cultures with frequent drinking and often less drinking per occasion, the lighter drinking occasions may be over-represented. If this is the case, we will expect that the difference in consumption volume between the Nordic and the Mediterranean countries is larger than observed in this study. Finally, we restricted the analyses to six countries, representing two different typologies of drinking patterns in Europe. Further studies of this topic are needed, and future analyses of data from other countries and other drinking cultures will likely add to and nuance the picture that emerged from this study.

## 5. Conclusions

Overall, aggregate-level findings suggest that declining trends in consumption volume are accompanied by similar trends in HED frequency. Individual-level findings suggest that among drinkers, the HED-oriented drinking culture in Nordic countries became less prominent over time along with the decline in consumption volume, while it remained fairly stable in the Mediterranean countries. Thus, the clear differences found between Nordic and Mediterranean countries at the start of the observation period may have weakened over the two decades from 1999 to 2019. More studies are needed to address the association between volume and pattern of drinking over time, also from other countries and drinking cultures. A better understanding of why consumption volume and HED frequency may go hand in hand will be important to inform policy making and prevention.

## Figures and Tables

**Figure 1 ijerph-19-07965-f001:**
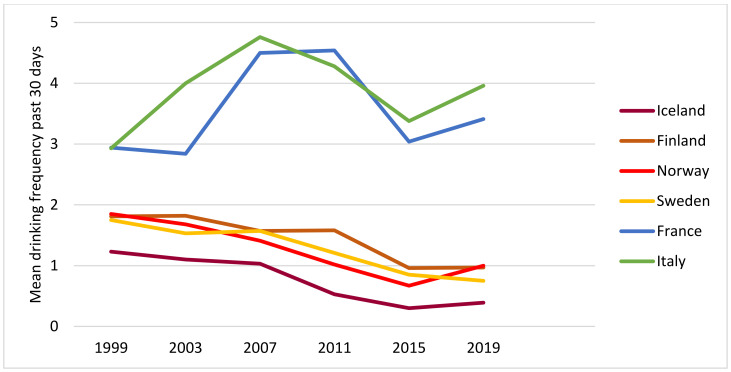
Mean drinking frequency past 30 days, by year and country. All students.

**Figure 2 ijerph-19-07965-f002:**
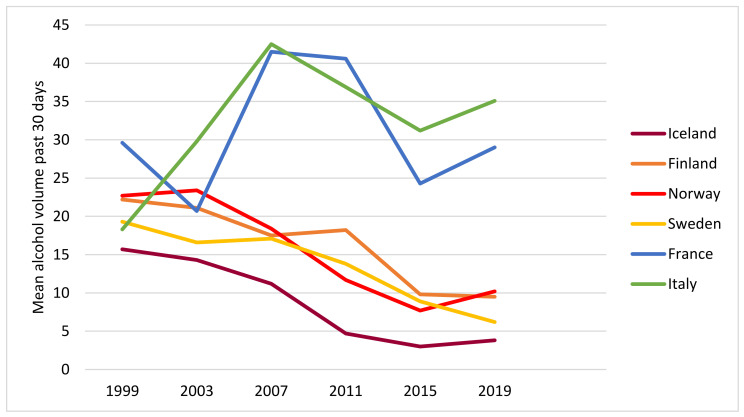
Mean alcohol volume past 30 days, by year and country. All students. Data for France in 2003 and 2011 were in part imputed.

**Figure 3 ijerph-19-07965-f003:**
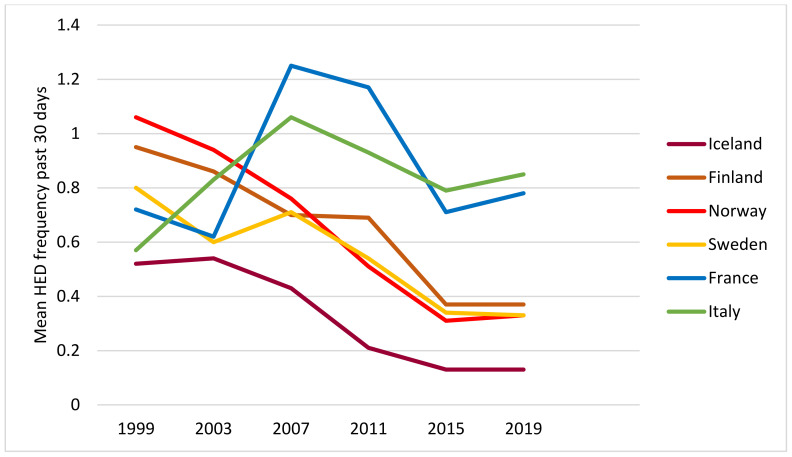
Mean HED frequency past 30 days by year and country. All students.

**Figure 4 ijerph-19-07965-f004:**
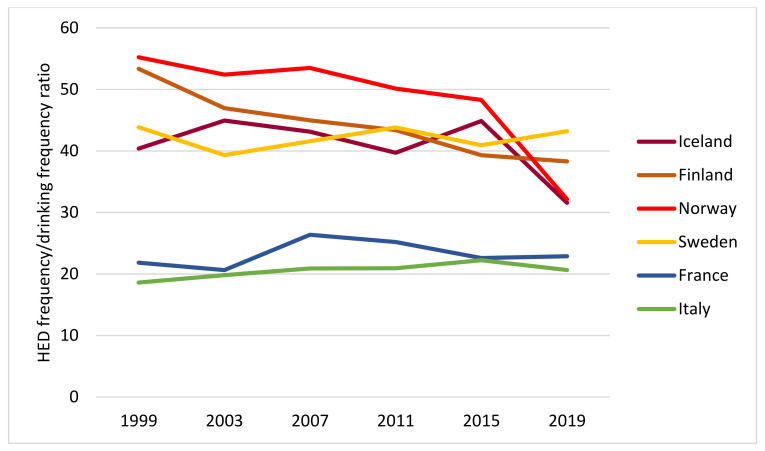
HED frequency/Drinking frequency ratio past 30 days, by year and country. All students.

**Figure 5 ijerph-19-07965-f005:**
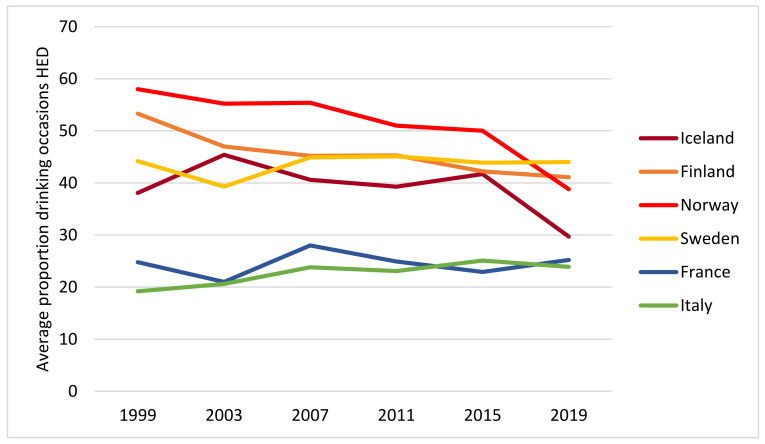
The average proportion of drinking occasions that were HED in the past 30 days, by year and country. Past 30 days drinkers only.

**Figure 6 ijerph-19-07965-f006:**
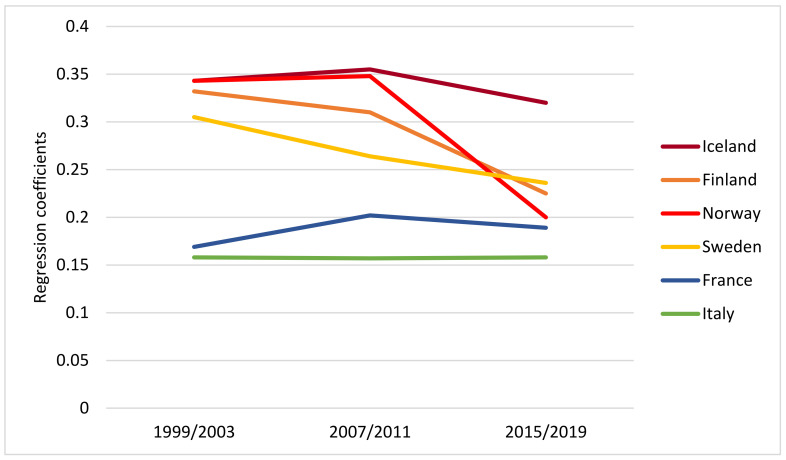
Regression coefficients for HED frequency on drinking frequency for the survey years 1999/2003, 2007/2011 and 2015/2019 by country.

**Table 1 ijerph-19-07965-t001:** The average frequency of drinking and HED, the ratio of HED:drinking frequency and consumption volume, all survey years by country, and all students [95% confidence intervals].

Country/Drinking Measures	Drinking Frequency	Volume	HED Frequency	HED/Drinking Frequency Ratio
Iceland	0.80 [0.77, 0.83]	9.2 [8.6, 9.8]	0.32 [0.30, 0.34]	0.40
Finland	1.42 [1.38, 1.46]	15.8 [15.1, 16.5]	0.63 [0.61, 0.65]	0.44
Norway	1.30 [1.26, 1.34]	16.0 [15.2, 16.8]	0.65 [0.63, 0.67]	0.50
Sweden	1.31 [1.27, 1.35]	14.1 [13.4, 14.8]	0.57 [0.55, 0.59]	0.44
France	3.59 [3.49, 3.69]	31.0 [29.6, 32.4]	0.89 [0.86, 0.92]	0.25
Italy	4.06 [3.98, 4.14]	34.2 [33.1, 35.3]	0.88 [0.86, 0.90]	0.22

Note: Data for France in 2003 and 2011 were in part imputed.

**Table 2 ijerph-19-07965-t002:** Proportion drinkers (past 30 days) by survey year and country.

Country/Survey Year	1999	2003	2007	2011	2015	2019
Iceland	42.4	36.0	31.3	17.1	9.3	11.2
Finland	60.7	53.6	47.9	47.8	32.1	30.8
Norway	53.8	48.4	41.8	34.6	23.7	25.5
Sweden	54.7	50.0	44.1	37.6	25.8	24.7
France	45.7	55.9	65.2	67.5	56.3	55.4
Italy	53.9	63.3	64.0	63.0	56.6	58.7

**Table 3 ijerph-19-07965-t003:** Correlations between mean HED frequency and two indicators of mean consumption volume by country. Aggregate data for the years from 1999 to 2019. Pearson’s correlation coefficient.

Country/Volume Measures	Calculated Volume	Drinking Frequency
Iceland	0.99 *p* < 0.001	0.99 *p* < 0.001
Finland	0.99 *p* < 0.001	0.98 *p* < 0.001
Norway	0.98 *p* < 0.001	0.98 *p* < 0.001
Sweden	0.97 *p* < 0.001	0.98 *p* < 0.001
France	0.97 *p* < 0.001	0.99 *p* < 0.001
Italy	0.98 *p* < 0.001	0.97 *p* = 0.002

Note: For France, in 2003 and 2011, data on volume was in part imputed.

**Table 4 ijerph-19-07965-t004:** HED frequency regressed on drinking frequency by survey year and country. Past 30-days drinkers only, weighted data. Regression coefficients and (SE). The 95% confidence intervals are in brackets.

Country/Year	1999/2003	2007/2011	2015/2019
Iceland	0.343	0.355	0.320
	(0.010)	(0.010)	(0.017)
	[0.322, 0.363]	[0.336, 0.373]	[0.287, 0.353]
Finland	0.332	0.310	0.225
	(0.008)	(0.006)	(0.007)
	[0.317, 0.347]	[0.298, 0.322]	[0.212, 0.238]
Norway	0.343	0.348	0.200
	(0.008)	(0.009)	(0.008)
	[0.329, 0.358]	[0.330, 0.366]	[0.183, 0.216]
Sweden	0.305	0.264	0.236
	(0.008)	(0.008)	(0.011)
	[0.290, 0.321]	[0.248, 0.280]	[0.214, 0.258]
France	0.169	0.220	0.189
	(0.005)	(0.005)	(0.005)
	[0.159, 0.180]	[0.211, 0.229]	[0.180, 0.198]
Italy	0.158	0.157	0.158
	(0.004)	(0.003)	(0.004)
	[0.151, 0.165]	[0.152, 0.163]	[0.150, 0.166]

## Data Availability

The ESPAD trend data are archived at the Italian National Research Council (CNR), where data can be applied for research purposes.

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
