# Peer review of "The Declining Trend in Adolescent Drinking: Do Volume and Drinking Pattern Go Hand in Hand?"

_ijerph, 2022, doi:10.3390/ijerph19137965_

Round 1
Reviewer 1 Report
There is a well done analysis of the observations alcohol drinking.
The problem of drinking alcohol is one of the main disorders of the risky lifestyle among young people. At the outset, it is necessary to very clearly emphasize his problem and the long-term effects of the later disease.
The discussion is comprehensive and deep, showing the authors' great erudition in this field. The conclusions were correctly drawn from the results. Most references are up to date and properly selected. However, I suggest a few corrections:
1. Please provide a definition of drinking alcohol (can be added in the introduction or in the discussion)?
2. I suggest observing other countries, especially in Eastern Europe, to better understand the situation in other parts of Europe.
3. Why in the summary the authors only mention the comments about the Nordic countries, and in France only Italy, without taking into account other high-consumption countries (Russia, Ukraine, Belarus or Australia or South America)?
4. Please explain to future readers what alcohol was considered for the analysis, as well as its frequency and quality.
5. So what interventions should be made, and even the observation methods should be maintained, while respecting the standards for the largest regions of the world, since the authors have already made a comparison between the countries represented?
6. Were any training courses conducted during the analysis and can the authors take into account sample educational programs in the indicated countries, both for adults and adolescents?
7. I suggest to check observations (line 61-63):
- Charkiewicz A.E et al. Changes in Dietary Patterns and the Nutritional Status in Men in the Metallurgical Industry in Poland Over A 21-Year Period. Ann Nutr Metab 2018;72:161-171. doi: 10.1159/000485389
- Conner KR, Abar B, Aldalur A, Chiang A, Hutchison M, Maisto SA, Stecker T. Alcohol-related consequences and the intention to seek care in treatment naïve women and men with severe alcohol use disorder. Addict Behav. 2022 Aug;131:107337. doi: 10.1016/j.addbeh.2022.107337.
- Soedamah-Muthu SS, Chaturvedi N, Fuller JH, et al; EURODIAB Prospective Complications Study Group: Do European people with type 1 diabetes consume a high atherogenic diet? 7-year follow-up of the EURODIAB prospective complications study. Eur J Nutr 2012; 52: 1701–1710
- Hulshof KF, Brussaard JH, Kruizinga AG, et al: Socio-economic status, dietary intake and 10 y trends: the Dutch National Food Consumption Survey. Eur J Clin Nutr 2003; 57: 128–137.
- Davies EL, Foxcroft DR, Puljevic C, Ferris JA, Winstock AR. Global comparisons of responses to alcohol health information labels: A cross sectional study of people who drink alcohol from 29 countries. Addict Behav. 2022 Aug;131:107330. doi: 10.1016/j.addbeh.2022.107330. Epub 2022 Apr 12. PMID: 35504111.
- Kim AJ, Sherry SB, Nealis LJ, Mushquash A, Lee-Baggley D, Stewart SH. Do symptoms of depression and anxiety contribute to heavy episodic drinking? A 3-wave longitudinal study of adult community members. Addict Behav. 2022 Jul;130:107295. doi: 10.1016/j.addbeh.2022.107295.
8. Please check the references section and remove the papers older than the year 2010.
9. Please correct a little bit the language by a native speaker.
Author Response
Thank you for the useful comments and suggestions. We have now revised the manuscript, taking into account the suggestions from the reviewer. A point-by-point response to Reviewer 1 follows below.
There is a well done analysis of the observations alcohol drinking.
The problem of drinking alcohol is one of the main disorders of the risky lifestyle among young people. At the outset, it is necessary to very clearly emphasize his problem and the long-term effects of the later disease.
RE: We agree that this important point was missing in the Introduction, and we have now added this as the very first part of the introduction (see page 1, line 30-32).
The discussion is comprehensive and deep, showing the authors' great erudition in this field. The conclusions were correctly drawn from the results. Most references are up to date and properly selected.
RE: We would like to thank Reviewer 1 for the positive comments regarding our manuscript.
However, I suggest a few corrections:
- Please provide a definition of drinking alcohol (can be added in the introduction or in the discussion)?
RE: We have added a sentence explaining how alcohol consumption/drinking behaviour is operationalized in the ESPAD surveys (see page 3, line 99-100).
- I suggest observing other countries, especially in Eastern Europe, to better understand the situation in other parts of Europe.
RE: While we agree it is clearly of interest to examine also other countries than the six included in our study, it was beyond the scope of this study. The present study employed a selection of countries, for which data are available from the ESPAD project, and which also represent two clearly different typologies of drinking cultures in Europe. This is stated in the Introduction (2nd para) and the Discussion (1st para), and is further emphasized in the Introduction of the revised version (page 2, line 48-49). We have now also added to the Limitations section in the Discussion, that further studies are needed, and that future analyses of data from other countries and drinking cultures will likely add to, and nuance, the picture (page 11 and 12, line 368-371). We have also paid attention to this issue in the Conclusion section (page 12, line 379-381).
- Why in the summary the authors only mention the comments about the Nordic countries, and in France only Italy, without taking into account other high-consumption countries (Russia, Ukraine, Belarus or Australia or South America)?
RE: We are not sure what the reviewer means by ‘summary’, however, we assume that ‘summary’ refers to the literature review in the Introduction. In order to motivate the study, we present two different typologies of drinking cultures; the one in the Nordic countries as opposed to that in southern European countries, including Italy and France. To clarify this point, we have expanded a bit on this part of the Introduction (page 2, line 48-49). Furthermore, we have also added to the Discussion a bit more information about typologies of drinking cultures and have emphasized the need for future studies from other countries and drinking cultures (page 11 and 12, line 368-371).
- Please explain to future readers what alcohol was considered for the analysis, as well as its frequency and quality.
RE: We have now added more details regarding measurement of alcohol consumption. That is, we have specified the types of alcoholic beverages (i.e., beer, cider, premixed drinks, wine and spirits) (page 3, line 99-100) and have stated that the alcohol content for each beverage type was used to calculate volume in centiliters of pure alcohol (page 3, line 115-122). The frequency of alcohol consumption was an overall measure of frequency in the past 30 days, as already described.
- So what interventions should be made, and even the observation methods should be maintained, while respecting the standards for the largest regions of the world, since the authors have already made a comparison between the countries represented?
RE: It is not entirely clear to us what the reviewer means by this comment, but we assume it applies to suggestions for future studies employing data from other countries than those included in our study. We have now added to the Discussion section, pointing out the need for further studies and using data from other countries than those included in the present study (page 11 and 12, line 368-371).
- Were any training courses conducted during the analysis and can the authors take into account sample educational programs in the indicated countries, both for adults and adolescents?
RE: We assume that the reviewer asks whether educational programs or other prevention measures (for adults and adolescents) were implemented in the countries over the observation period. We do not have systematic information in that regard, and it is likely very difficult, if at all possible, to obtain such information. For instance, in Norway, school education programs are delivered at the municipality level, and there is no monitoring at the national level about such interventions. For our study, it is of little importance to identify factors that contributed to the various trends in consumption in the six countries, as the study aim was to examine whether changes in consumption volume and heavy episodic drinking were correlated at the country level and whether individual-level associations differed between and within countries.
- I suggest to check observations (line 61-63):
- Charkiewicz A.E et al. Changes in Dietary Patterns and the Nutritional Status in Men in the Metallurgical Industry in Poland Over A 21-Year Period. Ann Nutr Metab 2018;72:161-171. doi: 10.1159/000485389
- Conner KR, Abar B, Aldalur A, Chiang A, Hutchison M, Maisto SA, Stecker T. Alcohol-related consequences and the intention to seek care in treatment naïve women and men with severe alcohol use disorder. Addict Behav. 2022 Aug;131:107337. doi: 10.1016/j.addbeh.2022.107337.
- Soedamah-Muthu SS, Chaturvedi N, Fuller JH, et al; EURODIAB Prospective Complications Study Group: Do European people with type 1 diabetes consume a high atherogenic diet? 7-year follow-up of the EURODIAB prospective complications study. Eur J Nutr 2012; 52: 1701–1710
- Hulshof KF, Brussaard JH, Kruizinga AG, et al: Socio-economic status, dietary intake and 10 y trends: the Dutch National Food Consumption Survey. Eur J Clin Nutr 2003; 57: 128–137.
- Davies EL, Foxcroft DR, Puljevic C, Ferris JA, Winstock AR. Global comparisons of responses to alcohol health information labels: A cross sectional study of people who drink alcohol from 29 countries. Addict Behav. 2022 Aug;131:107330. doi: 10.1016/j.addbeh.2022.107330. Epub 2022 Apr 12. PMID: 35504111.
- Kim AJ, Sherry SB, Nealis LJ, Mushquash A, Lee-Baggley D, Stewart SH. Do symptoms of depression and anxiety contribute to heavy episodic drinking? A 3-wave longitudinal study of adult community members. Addict Behav. 2022 Jul;130:107295. doi: 10.1016/j.addbeh.2022.107295.
RE: We have now examined carefully all the suggested references and whether they are relevant to our statement on lines 61 – 63 (i.e., “A growing literature shows that over the past two decades, prevalence of drinking as well as overall amount of drinking have declined among adolescents in many European countries»).
However, none of these references apply to the statement by examining trends in adolescent alcohol use or seem relevant otherwise, as described for each reference:
- Charkiewicz et al., 2018: this is a cohort study of adult men followed up from 1987 to 2010.
- Conner et al., 2022: this study examined alcohol-related consequences and intention to seek care among adults with severe alcohol use disorder.
- Soedamah-Muthu et al., 2013: this study examined nutrional intake among adults with Type 1 diabetes.
- Hulshof et al., 2003: this study examined diatary intake among adults.
- Davies et al., 2018: this study examined responses among adults (i.e. 16 – 85 years) to alcohol health information labels in a cross-sectional survey in 2018.
- Kim et al., 2022: this longitudinal cohort study of adults examined associations between heavy episodic drinking and symptoms of anxiety/depression.
- Please check the references section and remove the papers older than the year 2010.
RE: While we generally aim to provide recent literature, older references will sometimes be important or even essential to include. We have now checked each of the references older than 2010 carefully and searched literature databases for relevant publications that are more recent. Specifically:
- Mäkelä et al., 2006: this reference is now replaced by Babor et al., 2010.
- Cumming, 2009: This reference is a classic in statistical methods, it is still frequently cited, and it is kept as is.
- Room, 1992 and Olsson, 1990: these references are to explicitly stated ideas and cannot be substituted.
- Stockwell, et al., 2008: This is now replaced with a more recent publication (i.e., Stockwell et al., 2014).
- Please correct a little bit the language by a native speaker.
RE: Done.
Reviewer 2 Report
General impression
This paper aims to investigate the stability in drinking patterns within a population, irrespective of changes in the overall volume of consumption among youth. The data comes from the European School Survey Project on Alcohol and other Drugs (ESPAD); thus the age span is confined to those at age 16 or turning 16 during the survey year. More specifically the papers investigate; Norway, Finland, Iceland, and Sweden (Nordic countries) and France, Italy (Mediterranean countries), by utilizing the ESPAD data during the survey years: 1999, 2003, 2007, 2011, 2015, and 2019. The findings suggest that consumption volume and Heavy Episodic Drinking (HED) frequency decreased in the Nordic countries. A curvilinear trend was observed in France and Italy. In 1999/2003, the alcohol volume-HED correlation was significantly higher in the Nordic compared to the Mediterranean countries but became significantly weaker in Finland, Norway, and Sweden whereas it remained stable in France, Iceland, and Italy during the period. In several Nordic countries, the HED-oriented drinking culture appears to have weakened over time, along with the substantial decrease in adolescent drinking since 2000.
Overall, the paper is clear and well written, and the data used are appropriate for a topical question, namely the decline in alcohol consumption among adolescents. The merit with the paper is also that it draws on comparing different drinking cultures and how these have developed over time.
Minor revisions
1, The abstracts lack a sentence, or two, on a conclusion, please insert.
2, On page 2, lines 57-58, the authors use the term social forces, which in this context is a bit unlucky, since the two examples refer to age and SES, which I would not refer to as social forces. To me, a social force is something, an innovation, a shift in attitudes, or a movement that affects the entire, or most of a population. This is a bit picky but, to me, and here, the concept of social forces make me as a reader a bit uncertain of what the authors are trying to say.
3, In the discussion segment (p. 10, lines 280-303) the authors discuss possible explanations for the findings in the Nordic countries. These explanations all seem thought-provoking and reasonable. However, why are these explanations not valid in the Mediterranean countries as well, or rather why were not they affected by these as well? The authors need to address this in a sentence or two, since, as of now, these explanations seem a bit one-sided.
4, In the discussion segment, I would prefer if the authors, mentioned the underlying rationale for studying drinking patterns, namely the risk of health harms or social harm. It would be enough to mentions this, somewhere early on, say line 249. Just to remind the reader that adolescent drinking is not a “puzzle” but may be associated to acute and/or long-term consequences.
5, The paper compares different countries, and drinking cultures, and as such it is hard to address implications. Still, it would be appreciated if the conclusion mentioned either implication for policy, practitioners, or prevention.
Author Response
Thank you for the useful comments and suggestions. We have now revised the manuscript, taking into account the suggestions from the reviewer. A point-by-point response to Reviewer 2 follows below.
General impression
This paper aims to investigate the stability in drinking patterns within a population, irrespective of changes in the overall volume of consumption among youth. The data comes from the European School Survey Project on Alcohol and other Drugs (ESPAD); thus the age span is confined to those at age 16 or turning 16 during the survey year. More specifically the papers investigate; Norway, Finland, Iceland, and Sweden (Nordic countries) and France, Italy (Mediterranean countries), by utilizing the ESPAD data during the survey years: 1999, 2003, 2007, 2011, 2015, and 2019. The findings suggest that consumption volume and Heavy Episodic Drinking (HED) frequency decreased in the Nordic countries. A curvilinear trend was observed in France and Italy. In 1999/2003, the alcohol volume-HED correlation was significantly higher in the Nordic compared to the Mediterranean countries but became significantly weaker in Finland, Norway, and Sweden whereas it remained stable in France, Iceland, and Italy during the period. In several Nordic countries, the HED-oriented drinking culture appears to have weakened over time, along with the substantial decrease in adolescent drinking since 2000.
Overall, the paper is clear and well written, and the data used are appropriate for a topical question, namely the decline in alcohol consumption among adolescents. The merit with the paper is also that it draws on comparing different drinking cultures and how these have developed over time.
RE: We would like to thank Reviewer 2 for the positive comments regarding our manuscript.
Minor revisions
1, The abstracts lack a sentence, or two, on a conclusion, please insert.
RE: We have now added and rephrased the conclusion part of the abstract (page 1, line 21-24).
2, On page 2, lines 57-58, the authors use the term social forces, which in this context is a bit unlucky, since the two examples refer to age and SES, which I would not refer to as social forces. To me, a social force is something, an innovation, a shift in attitudes, or a movement that affects the entire, or most of a population. This is a bit picky but, to me, and here, the concept of social forces make me as a reader a bit uncertain of what the authors are trying to say.
RE: We agree, this is a good point. The term ‘social forces’ have now been replaced with ‘social factors’ (page 2, line 66).
3, In the discussion segment (p. 10, lines 280-303) the authors discuss possible explanations for the findings in the Nordic countries. These explanations all seem thought-provoking and reasonable. However, why are these explanations not valid in the Mediterranean countries as well, or rather why were not they affected by these as well? The authors need to address this in a sentence or two, since, as of now, these explanations seem a bit one-sided.
RE: We agree, and we have added a short paragraph on this (page 11, line 325-334).
4, In the discussion segment, I would prefer if the authors, mentioned the underlying rationale for studying drinking patterns, namely the risk of health harms or social harm. It would be enough to mentions this, somewhere early on, say line 249. Just to remind the reader that adolescent drinking is not a “puzzle” but may be associated to acute and/or long-term consequences.
RE: We agree and have added a sentence addressing this issue in the very beginning of the discussion section (page 10, line 266-268).
5, The paper compares different countries, and drinking cultures, and as such it is hard to address implications. Still, it would be appreciated if the conclusion mentioned either implication for policy, practitioners, or prevention.
RE: We have added two sentences to the Conclusion section about possible implications for policy making and prevention (page 12, 381-382).
Reviewer 3 Report
The current manuscript aimed to investigate the potential association between the alcohol drinking pattern (in terms of heavy episodic drinking, HED) and the declined drinking amount among adolescents in European countries especially in Nordic countries that represent an ‘intoxication-prone’ drinking culture as opposed to the Mediterranean countries that represent a ‘moderation-prone’ drinking culture.
The study utilized dataset from the European School Survey Project on Alcohol and other Drugs (ESPAD) and demonstrated clear distinction between Nordic countries and Mediterranean countries (less vs. more total volume, more vs. less HED/drinking frequency ratio), clear associations between the HED frequency with both drinking volume and drinking frequency.
While the study is interesting providing data on different aspects of alcohol consumption (overall amount and HED frequency), there are some concerns/comments/questions listed below:
L99-L101: How were these responses recoded? Some details would be helpful. For example, the authors explained the second “30” in the parenthesis, however, how was the first “30” obtained? Shouldn’t it be “29.5”?
L118-L121: Similar as the above comment, how were these “recoded”?
L152-L153: Please describe with more details on how this ‘proxy measure for volume’ was derived.
L169: “Substantial differences” is arbitrary, this statement can be better supported with proper statistical tests (e.g., t-test or ANOVA).
Figures: Figures of the manuscript need to be clearly labeled (e.g., what are the y-axis?). Also for each time point, a measure of variance should be presented along with the mean (e.g., SD or SE).
L218: The text here indicates: the decreasing trend over time is “more so in Finland and Norway”, however, from Figure 4 it’s hard to tell.
Table 4: Why using regression analysis here? Based on the text above the table (L230-L232), this was to estimate the association between HED frequency and the overall drinking frequency. Correlation analysis seems to fit better the purpose of expressing association between two variables here.
Author Response
Thank you for the useful comments and suggestions. We have now revised the manuscript, taking into account the suggestions from the reviewer. A point-by-point response to Reviewer 3 follows below.
The current manuscript aimed to investigate the potential association between the alcohol drinking pattern (in terms of heavy episodic drinking, HED) and the declined drinking amount among adolescents in European countries especially in Nordic countries that represent an ‘intoxication-prone’ drinking culture as opposed to the Mediterranean countries that represent a ‘moderation-prone’ drinking culture.
The study utilized dataset from the European School Survey Project on Alcohol and other Drugs (ESPAD) and demonstrated clear distinction between Nordic countries and Mediterranean countries (less vs. more total volume, more vs. less HED/drinking frequency ratio), clear associations between the HED frequency with both drinking volume and drinking frequency.
While the study is interesting providing data on different aspects of alcohol consumption (overall amount and HED frequency), there are some concerns/comments/questions listed below:
L99-L101: How were these responses recoded? Some details would be helpful. For example, the authors explained the second “30” in the parenthesis, however, how was the first “30” obtained? Shouldn’t it be “29.5”?
RE: The first 30 is the integer mid-point for the category. Thus, two categories took the value 30. A very small fraction of the students ticket off for the 40+ category (i.e. less than 1 %), which is now stated.
L118-L121: Similar as the above comment, how were these “recoded”?
RE: We have now expanded the description and avoided the term ‘recode’. The sentences now read as follows: Response options were ‘none’, ‘1 time’, ‘2 times’, ‘3–5 times’, ‘6–9 times’, ‘10 or more times’ and by employing mid-point values, these response categories took the values ‘0’, ‘1’, ‘2’, ‘4’, ‘7.5’, and ‘12’, respectively. For some respondents, who reported drinking frequency ‘1 – 2 times’ and HED frequency ‘2 times’, the latter took the value ‘1.5 times’.
L152-L153: Please describe with more details on how this ‘proxy measure for volume’ was derived.
RE: We applied the frequency measure as an indicator (or proxy measure) of volume. The frequency measure is described in the Measures section. The reasons why drinking frequency may serve as an indicator – or proxy – for volume are stated in the preceding paragraph.
L169: “Substantial differences” is arbitrary, this statement can be better supported with proper statistical tests (e.g., t-test or ANOVA).
RE: We have now included 95 CIs in Table 1. Non-overlapping confidence intervals are statistically different.
Figures: Figures of the manuscript need to be clearly labeled (e.g., what are the y-axis?). Also for each time point, a measure of variance should be presented along with the mean (e.g., SD or SE).
RE: Labels are now included in the figures. While variance measures (e.g., +/- 2 SE) could, in principle, be presented, this would imply an additional 72 data points to each figure, which would clearly hamper readability.
L218: The text here indicates: the decreasing trend over time is “more so in Finland and Norway”, however, from Figure 4 it’s hard to tell.
RE: We have now removed this part of the sentence.
Table 4: Why using regression analysis here? Based on the text above the table (L230-L232), this was to estimate the association between HED frequency and the overall drinking frequency. Correlation analysis seems to fit better the purpose of expressing association between two variables here.
RE: We acknowledge that we should have pointed out why we employed linear regression analysis. There are some important differences between correlation analysis and bivariate linear regression: a correlation analysis is symmetrical (i.e., the correlation between x and y is the same as the correlation between y and x), whereas linear regression analysis will provide different measures of association depending on whether y is regressed on x, or x is regressed on y. Let us, as an example, consider estimates from Sweden for the years 1999/2003 in our data set: the linear regression coefficient for HED frequency regressed on drinking frequency is 0.305, whereas the regression coefficient for drinking frequency regressed on HED frequency is 1.191. The corresponding figures for Italy is 0.158 and 1.741. A linear regression analysis predicts a change in y, given a one unit change in x (the slope of the regression line), and it estimates an intercept. Thus, in our context, we analyse how much a change in drinking frequency predicts a change in HED frequency (the regression coefficient), that is, we regressed HED frequency on drinking frequency. A high regression coefficient is consistent with a drinking culture where an increase in drinking frequency to a large extent implies an increase in HED frequency. We have now added a short explanation in the Methods section (page 4, line 171-172).